# Changes in the Bacterioplankton Community Structure from Southern Gulf of Mexico During a Simulated Crude Oil Spill at Mesocosm Scale

**DOI:** 10.3390/microorganisms7100441

**Published:** 2019-10-11

**Authors:** Sonia S. Valencia-Agami, Daniel Cerqueda-García, Sébastien Putzeys, María Magdalena Uribe-Flores, Norberto Ulises García-Cruz, Daniel Pech, Jorge Herrera-Silveira, M. Leopoldina Aguirre-Macedo, José Q. García-Maldonado

**Affiliations:** 1Centro de Investigación y de Estudios Avanzados del Instituto Politécnico Nacional, Mérida, Yucatán 97310, Mexico; sonia.valencia@cinvestav.mx (S.S.V.-A.); maria.uribe@cinvestav.mx (M.M.U.-F.); jorge.herrera@cinvestav.mx (J.H.-S.); 2Consorcio de Investigación del Golfo de México (CIGoM)—Centro de Investigación y de Estudios Avanzados del Instituto Politécnico Nacional, Mérida, Yucatán 97310, Mexico; dacegabiol@ciencias.unam.mx (D.C.-G.); sebastien.putzeys@cinvestav.mx (S.P.); norbertoulisesg@gmail.com (N.U.G.-C.); 3Laboratorio de Biodiversidad Marina y Cambio Climático, El Colegio de la Frontera Sur, Lerma Campeche, Campeche 24500, Mexico; dpech@ecosur.mx; 4CONACYT – Centro de Investigación y de Estudios Avanzados del Instituto Politécnico Nacional, Mérida, Yucatán 97310, Mexico

**Keywords:** light crude oil, bacterioplankton, southern Gulf of Mexico, 16S rRNA, mesocosm experiment

## Abstract

The southern Gulf of Mexico (sGoM) is highly susceptible to receiving environmental impacts due to the recent increase in oil-related activities. In this study, we assessed the changes in the bacterioplankton community structure caused by a simulated oil spill at mesocosms scale. The 16S rRNA gene sequencing analysis indicated that the initial bacterial community was mainly represented by Gamma-proteobacteria, Alpha-proteobacteria, Flavobacteriia, and Cyanobacteria. The hydrocarbon degradation activity, measured as the number of culturable hydrocarbonoclastic bacteria (CHB) and by the copy number of the *alkB* gene, was relatively low at the beginning of the experiment. However, after four days, the hydrocarbonoclastic activity reached its maximum values and was accompanied by increases in the relative abundance of the well-known hydrocarbonoclastic *Alteromonas*. At the end of the experiment, the diversity was restored to similar values as those observed in the initial time, although the community structure and composition were clearly different, where *Marivita*, *Pseudohongiella*, and *Oleibacter* were detected to have differential abundances on days eight–14. These changes were related with total nitrogen (*p* value = 0.030 and *r*^2^ = 0.22) and polycyclic aromatic hydrocarbons (*p* value = 0.048 and *r*^2^ = 0.25), according to PERMANOVA. The results of this study contribute to the understanding of the potential response of the bacterioplankton from sGoM to crude oil spills.

## 1. Introduction

The Gulf of Mexico (GoM) is a hotspot of biological diversity in which several oil-related industries are hosted, representing a high risk for oil spill disasters [1,2]. Two of the most important precedents of disaster were the Deepwater Horizon massive oil spill (DWH), which occurred in waters off Louisiana, USA during 2010 [3], in which more than 700,000 tons of crude oil was spilled into the GoM [4] and the Ixtoc-I oil spill during 1979, with more than 3.4 million barrels of crude oil spilled into the southern Gulf of Mexico (sGoM) for nine months [5].

Microbial communities have different responses to environmental perturbations [6,7] depending on the local environmental conditions (e.g., temperature, nutrients, oxygen, and salinity) and the duration of the disturbance [7]. In particular, for the northern GoM, several studies were conducted to detect changes in the bacterioplankton community caused by the DWH disturbance. It was observed that the microbial community responded to hydrocarbons only a few weeks after the oil spill, and it was related with the long-term adaptation to natural oil released into the environment [8,9,10,11,12]. Moreover, a dominance of microorganisms related with the consumption of aliphatic hydrocarbons was found at the initial stage of the spill (e.g., Oceanospirillales and *Pseudomonas*), while potential degraders of aromatic compounds (e.g., *Colwellia* and *Cycloclasticus*) were observed in a second stage [8,11,13]. Towards the end of the DWH spill, another shift in the microbial community was detected, where microorganisms involved in the degradation of complex organic matter as methylotrophs (e.g., Flavobacteriia, Alteromonadaceae, and Rhodobacteraceae) were dominant [14].

For over a decade, several simulations at mesocosm scale have been performed for the evaluation of the effect of hydrocarbons on planktonic microorganisms [15,16]. At this scale, it has been evidenced that low molecular weight polycyclic aromatic hydrocarbons (PAHs) can be degraded in a period of two days by several microorganisms [17]. The genus *Cycloclasticus* has been reported as one of the most important degraders of low molecular weight PAHs in the cold waters of Galicia, Spain [18]. Moreover, previous investigations at mesocosms scale have reported *Alcanivorax* as relevant microorganisms for cleaning oil-polluted marine systems, due to their high capability to use aliphatic compounds as carbon source [19,20,21]. For northern GoM, a rapid formation of microbe-oil aggregates and significant changes in the community composition from coastal surface water, after exposure to oil and Corexit dispersant, have been previously detected by mesocosms experiments. This rapid response of the microbial community was related to the presence of natural oil seepages in the GoM, and hence, the resident microbial communities are well-adapted to the presence of hydrocarbon in the environment [22].

The southern Gulf of Mexico is considered the largest base of the petroleum industry in Mexico, and explorations have been performed since 1970 and continue to increase until current times [5,23]. Despite several studies having been conducted after the Ixtoc-I disaster, the environmental alterations caused by this blowout remain scarcely assessed. Thus, contributing with the understanding of the potential response of the bacterioplankton from sGoM to crude oil spills, this study aimed to assess the effect of light crude oil over the native microbial community from sGoM, through a simulated oil spill under a mesocosm experiment.

## 2. Materials and Methods

### 2.1. Sampling and Set-Up of the Mesocosm System

Superficial seawater (< 3 m depth) was collected during October 2017 at Progreso harbor, in the Yucatan Peninsula, Mexico (21°20” N; 89°40” W). A single tank (2800 L total capacity; 1.75 m high; 1.55 m diameter) was filled up to 2500 L with seawater and transported to the CINVESTAV-IPN facilities in Merida, Yucatan. The mesocosm was equipped with an electric thruster to homogenize seawater temperature, oxygen, and salinity (Scheme 1). Before filling the tank, natural seawater was filtered through a nylon mesh (200 µm) to remove mesoplankton and detritus. An acclimatization period of 24 h was considered in this study to allow the stabilization of the microbial communities to the experimental conditions. The experiment was performed outdoors with natural sunlight cycles during 15 days, with constant stirring at low speed (0.125 m s^−1^), and no nutrient addition was performed. For the determination of the microbial community composition in the seawater acclimatized to the mesocosm before crude oil addition, a sample was collected and catalogued as “Reference” (R). At day zero, the tank was supplemented with light crude oil (Ixtoc-I well; 31–39°API) at 80 mg^−1^ L.

During the experiment, one-meter depth water samples were collected from the mesocosm on days zero, two, four, six, eight, 10, and 14 to measure oxygen, salinity, temperature, nutrients, and hydrocarbons. The Winkler method was approached according to the steps suggested by Carrit and Carpenter [24] and Bryan et al. [25] for oxygen measurements. The entire content of each Winkler bottle was titrated manually during ~3 min by colorimetric end-point detection [26]. A YSI probe was used to obtain measurements of salinity and temperature from seawater samples. For the inorganic nutrients, seawater samples were collected using sterile 1 L bottles and stored at −20 °C until analysis. Nitrate plus nitrite (NO_3_^−^+NO_2_^−^), nitrite (NO^2-^), orthophosphate (PO_4_^3−^), and silicic acid (SiO_4_^4−^) analyses were performed according to Strickland and Parsons [27], using an Agilent (Cary 60 UV-VIS) spectrophotometer. The detection limits for each inorganic nutrient were as follows: 0.03 µM for nitrates, 0.01 µM for nitrites, 0.02 µM for phosphates, and 0.05 µM for silicates. Total Nitrogen and carbon were quantified with 50 µL seawater samples using a Thermo Scientific FLASH 2000 CHNS/O analyzer.

### 2.2. Quantification of Hydrocarbons in the Water Column

Crude oil concentration was measured as the total petroleum hydrocarbon (TPH). It was determined as the sum of aliphatic, aromatics (PAHs) and unresolved complex mixture (UCM) concentrations. Hydrocarbons were obtained by liquid–liquid extraction with 150 mL of dichloromethane and 4 L of one-meter depth seawater from the mesocosm. PAHs and UCM fractions were analyzed by high-resolution Gas Chromatography-Mass Spectroscopy (GC-MS) Perkin Elmer Clarus 500 and Gas Chromatography with Agilent 7890A (Santa Clara, CA, USA) equipped with a flame ionization detector (GC-FID) for aliphatic fraction.

### 2.3. Estimation of Culturable Hydrocarbonoclastic Bacteria and Quantification of the alkB Gene

Water samples (15 mL centrifugate tubes) were collected specifically for assessing the culturable hydrocarbonoclastic bacteria (CHB). Estimates of the number of CHB were obtained using the most probable number (MPN) method reported by [23,28]. The determination was performed using sterile tubes (10 mL), each containing 5 mL of sterile Bushnell-Hass (B-H) medium (Difco, Livonia, Michigan, USA) conditioned with 2% NaCl, pH 7. Test tubes were inoculated with seawater from the mesocosm, light crude oil (0.178 mg L^−1^) as the sole carbon source, and resazurin as an indicator of bacterial growth. The quantification of the *alkB* gene, related to first-step hydroxylases involved in the metabolism of alkanes, was performed on a Rotor-Gene Q System (Qiagen, Hilden, Germany) with the primers and the protocol previously reported by Uribe-Flores et al. [29], with coefficient of determination (*r*^2^) of 0.98 and PCR amplification efficiency of 98.3%.

### 2.4. Bacterial Community Analyses

Changes in the microbial community were followed by 16S rRNA Illumina sequencing. DNA extraction was performed using the PowerWater Sterivex DNA Isolation kit (Mo Bio Laboratories, Inc.), according to manufacturer’s instructions. Seawater from the mesocosm was filtered (5 L) through a 0.22 µm Sterivex cartridge filter (Millipore Corp., Bedford, MA, USA) using a peristaltic pump. The quality of DNA extraction was evaluated by agarose gel (1%) electrophoresis. Amplification of hypervariable V3 and V4 region of 16S rRNA was performed using the primers and the conditions suggested by Klindworth et al. [30]. Amplicons were purified with AMPure XP beads and indexed using the Nextera XT kit, according to the library preparation protocol recommended by the manufacturer. Indexed PCR products were purified and quantified with a Qubit^®^ 3.0 Fluorometer using the Qubit dsDNA HS Assay Kit (Life Technologies, Carlsbad, CA, USA). Amplicon size was verified by capillary electrophoresis at QIAxcel Advanced (QIAGEN, Valencia, CA, USA). Individual amplicons were diluted in 10 mM Tris (pH 8.5) and pulled at equimolar concentrations (4 nM). Sequencing was carried out in CINVESTAV-Mérida using an Illumina-Miseq platform (Illumina, San Diego, CA, USA), with the MiSeq reagent kit V3 (2 × 300), following the manufacturer’s recommendations.

### 2.5. Bioinformatics and Statistics

The demultiplexed paired-end reads (2x300), in the fastq format were processed with the QIIME2 pipeline (2018.8) [31]. The error correction and denoising to resolve the amplicon sequence variants (ASVs) were performed with the DADA2 plugin [32,33]. The length of sequences was 250 bp after trimming. Removing chimeras with the “consensus” method. The representative ASVs were taxonomically assigned with the V-SEARCH consensus taxonomy classifier plugin [34] using the SILVA database (v.128) as a reference. A phylogenetic tree was built with the reference ASVs with the FastTree algorithm [35]. The samples were rarefied at a sample depth of 30,000 reads, then transformed to percentage to obtain relative abundances. The abundance table was exported to the R environment, and the statistical analysis and visualization was performed with the phyloseq [36], vegan [37], and ggplot2 [38] libraries. Pairwise dissimilarities were calculated using UniFrac (weighted) metrics [39]. Each resulting dissimilarity matrix was used to visualize differences in the samples through principal coordinates ordination analysis (PCoA), using the “phyloseq” packages in R [40]. The data matrix was constructed with the means of each variable per day and was used as independent variables. To test significant differences between the discriminated groups according to sampled day and physiochemical parameters, similarities were analyzed by global and pairwise PERMANOVA tests, using “adonis” function in “vegan” [39]. *p* values were obtained using 999 permutations. Shannon-Weaver diversity index was estimated in vegan library. A Kruskal Wallis test was performed to detect ASVs with differential abundance among three groups of samples based on changes in the Shannon index: Group 1 (samples R and zero), Group 2 (samples two, four, and six), and Group 3 (samples eight, 10, and 14). The taxonomic assignation of the differential ASVs, was improved, with the selection of the three best blast hits from the RefSeq NCBI database. Raw sequence data produced in this study was deposited in NCBI under the Bioproject accession number PRJNA560042

## 3. Results

### 3.1. Physicochemical Analyses

All the physicochemical parameters measured through the 15 days of the mesocosms experiment are shown in Table 1. The temperature ranged from 26.30–28.30 °C, while the initial value for oxygen was 1.90 mg L^−1^, and the final concentration was 4.54 mg L^−1^ at day 14. Salinity ranged from 35.53–36.70 PSU. Inorganic nutrients (PO_4_, NO_2_, NO_3_, SiO_4_, and NH_4_), Total C, and Total N, had an increase from day zero to day 14, however, this increment was more conspicuous in the nitrogen concentration range (0.04 to 1.55%).

### 3.2. Estimation of the Bacterial Hydrocarbon Degrading Activity

The initial values (day zero) of TPH concentration, CHB count, and *alkB* gene copies were 31.96 µg L^−1^, 70 CFU, and 2 × 10^5^ copies µL^−1^, correspondingly. At day four, increases in these measurements were observed (TPH = 6,503.87 µg L^−1^, CHB = 23,000, and *alkB* gene = 1.2 × 10^6^ copies µL^−1^). However, at day six the TPH concentration (117.48 µg L^−1^) and *alkB* gene copies (5 × 10^4^ copies µL^−1^) decreased considerably. At the end of the experiment the values of TPH concentration and the *alkB* gene copies remained low (595.03 µg L^−1^ and 7 × 10^3^ copies µL^−1^, correspondingly); additionally, the CHB count decreased to 90 CFU (Figure 1).

### 3.3. Bacterial Community Structure and Composition

Initial bacterioplankton community displayed the highest values of Shannon-Weaver index (Table 2). PCoA indicated that community structure after the oil addition was highly distant from the initial time (Figure 2). In terms of bacterial composition, the most abundant groups at the beginning of the experiment (day zero) were Gamma-proteobacteria, Alpha-proteobacteria, Verrucomicrobiae, Flavobacteriia, Cyanobacteria, and Sphingobacteriia (Figure 3a). During days two to four, Shannon-Weaver diversity (H’) ranged from 4.66 to 4.74 (Table 2), and Alpha- and Gamma-proteobacteria reached their maximum relative abundances, respectively (Figure 3a). At a lower taxonomical level, *Alteromonas* increased considerably at day four. By the last period of the experiment (days eight to 14), the values of the diversity index were similar to those found at the initial time (Table 2). However, the final community composition was different, with notorious increases in the relative abundance of Sphingobacteriia and Bacteroidetes *Incertae Sedis* at the last day of the experiment. At the genus level, *Marivita* and *Phaeodactillibacter* showed an increment in their relative abundances from day eight to 14 (Figure 3a,b). Genera with low relative abundance (<3% and classified as others) represented 40–60% of the total bacterioplankton composition throughout the experiment (Figure 3b). According to PERMANOVA, changes detected in the bacterial community were associated with the concentration of the aromatic fraction (Appendix A) and total nitrogen concentration dissolved in the water column (Table 3).

### 3.4. Differential Abundance Analysis

The differential ASVs belonging to *Candidatus Thiobios*, *Candidatus Actinomarina*, *Acinetobacter*, and *Synechococcus* were characteristic of days R and zero (Figure 4), while several unassigned ASVs were differentiated from initial time to day six. BLAST analyses from these unassigned ASVs, indicated that the sequences were related with the genera *Lewinella* (92.5%), *Prochlorococcus* (96.37%), *Angustibacter* (96.80%), *Ilumatobacter* (97.2), and *Candidatus Pelagibacter* (96.78%) (Appendix A). *Marinobacter* was detected with differentiated abundances from the beginning of the experiment until day eight. From day eight until the end of the experiment, *Marivita* and *Pseudohongiella* were conspicuous (Figure 4).

## 4. Discussion

The mesocosm design used for this research allowed to maintain similar environmental conditions to those previously reported for the superficial seawater from sGoM. The temperature, oxygen, salinity, and nutrient concentration measured during the mesocosm experiment were within the ranges previously reported for the water column of the Yucatan Peninsula [41,42]. The initial community composition of this study was comparable to the typical surface marine bacterioplankton, consisting of Alpha-proteobacteria, Gamma-proteobacteria, Flavobacteriia, and Cyanobacteria, as dominant members [43]. Thus, the results obtained from this study represent a proxy of the response of the native bacterioplankton community from sGoM to oil spills.

### 4.1. Shifts in the Bacterioplankton Community after the Simulated Oil Spill

In agreement with previous studies significant changes in the bacterial community were observed were observed after 48 h of exposition to crude oil [44,45,46]. During this time, considerable increases in the relative abundances of Alpha-proteobacteria and decreases in Cyanobacteria and Gamma-proteobacteria were observed (Figure 3b). These results differ from previous works in which increases in the relative abundances of Gamma-proteobacteria and Clostridia, have been observed during the initial period of expositions to contaminants in sites with low temperature [47,48]. We hypothesized that decreases in the relative abundances of Cyanobacteria could be related with the direct exposition to oil hydrocarbons, however the reduction in the amount of available light in the water column caused by the oil film on the surface, could be also associated with the affectation in the photosynthetic activity of Cyanobacteria.

Previous studies have reported a potential recovery of the bacterioplankton diversity in late periods of simulated hydrocarbon spills [49]. In concordance with this study, the diversity at the end of our experiment returned to similar values as those observed in the initial time. However, the bacterial composition was different from the initial community, which also in agreement with previous studies where it has been suggested that communities do not fully recover from the disturbance and move to an alternative stable state, where they can maintain their functional capacity [7,50,51].

In our study, some of the main changes in the bacterial composition after the simulated oil spill were related with decreases in the relative abundances of *Candidatus Thiobio*, *Candidatus Actinomarina*, *Acinetobacter,* and *Synechococcus*. Previous studies have reported to *Synechococcus* and *Actinomarina* as sensitive taxa for contaminated sites with heavy metals [52]. However, futures studies are needed in order to understand the oil effect over these genera.

Several works have demonstrated that changes in the bacterial community can be largely attributed to specific physicochemical parameters and hydrocarbons [53,54]. In this study, changes in the bacterioplankton structure were related to total nitrogen and the aromatic compounds concentrations, based on PERMANOVA analysis (Table 3). Accordingly, nitrogen had been related with the microbial growth and is commonly catalogued as limiting nutrient during crude oil degradation in natural environments [55]. Moreover, the persistence of aromatic hydrocarbons in the environment due to their low solubility have been well recognized to have influence on the bacterioplankton by previous studies [56,57,58].

### 4.2. Hydrocarbonoclastic Bacterial Activity

Previous mesocosm studies that have undertaken lower temperatures (≤18 °C) have found that the main crude oil removal occurred from 15 to 20 days [21,59]. Unlikely, in this study the highest degradation activity was observed at day four, which highlight the importance of the seawater temperature for hydrocarbon degradation activity as previously was evidenced by Brackstad et al. [60].

During the period of highest crude oil degradation (day two to four), increases in the relative abundance of the *Alteromonas*, *Haroferula*, *Rugeria*, and *Alcanivorax* were distinctive (Figure 3b). These genera have been previously recognized to be involved in the degradation of aromatic and aliphatic hydrocarbons [61,62,63,64,65,66,67,68]. Therefore, we considered that these microorganisms have been related to the consumption of hydrocarbons in our experiment and may be explored for future bioremediation studies.

By the last stage of the experiment (days eight–14), increases in the relative abundances of Sphingobacteriia were observed (Figure 3a). Moreover, ASVs of *Marivita* and *Oleibacter* presented differential abundance in this period of the experiment (Figure 4). Previous studies have clearly showed that members of Sphingobacteriia can be n-alkane-degrading specialists [69]. *Marivita* belongs to the Roseobacter lineage (Alpha-proteobacteria) and has been reported to degrade aromatic and aliphatic compounds [64,70,71]. The genus *Oleibacter* has been reported with the capacity to degrade aliphatic hydrocarbons in tropical environments [72,73,74]. Based on the information presented above, we assume that Sphingobacteriia, *Marivita*, and *Oleibacter* had important participation related with the hydrocarbon degradation during the last days of our mesocosm system, however further research is required for a better understanding of its contribution in the microbial community.

## 5. Conclusions

The mesocosm experiment of this study evidenced a decrease in bacterioplankton diversity and changes in the community composition from sGoM after two days from the simulated oil spill. Increases in the relative abundances of Alpha-proteobacteria and decreases in Cyanobacteria and Gamma-proteobacteria were some of the main changes in the bacterial composition observed in this period of the experiment. According to the TPH concentration, counts of culturable hydrocarbonoclastic bacteria, and *alkB* gene copy number, the greatest hydrocarbon degradation activity performed by the bacterioplankton of sGoM occurred after four days from the disturbance. *Alteromonas*, *Haloferula*, *Ruegiera*, and *Alcanivorax* were found as the main groups putatively involved in the hydrocarbon degradation, based on increases in their relative abundances at this point of the experiment. Despite of the bacterial diversity being recovered at the end of the experiment (14 days), the community composition was different from the initial time. To our knowledge, this is the first mesocosm experiment to evaluate the effect of oil spills over the bacterioplankton community from sGoM. All the results presented here can be employed as a guideline for future studies aiming to evaluate bioremediation strategies for marine environments.

## Figures and Tables

**Scheme 1 microorganisms-07-00441-sch001:**
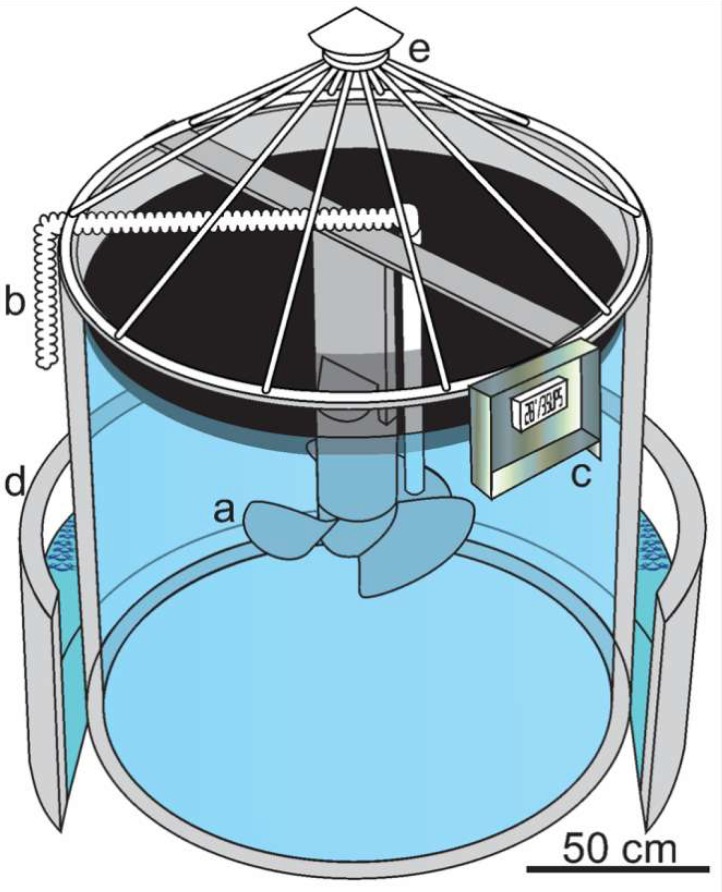
Mesocosm design; (**a**) electric thruster, (**b**) plastic siphon for sampling, (**c**) remote control of salinity and temperature sensors, (**d**) water bath as cooler system, and (**e**) plastic coating.

**Figure 1 microorganisms-07-00441-f001:**
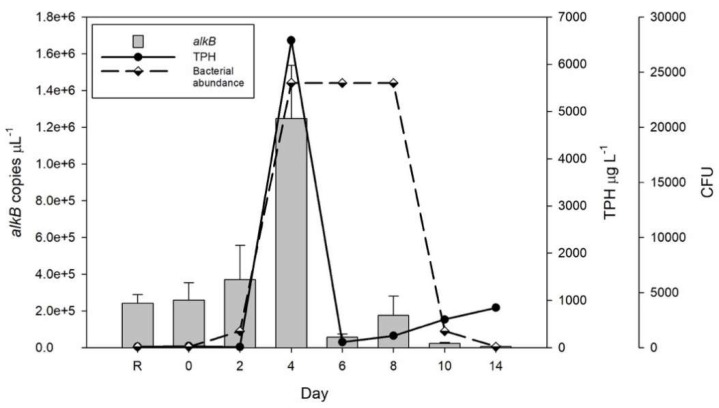
Abundances of culturable hydrocarbonoclastic bacteria, concentration of total petroleum hydrocarbons (TPH), and copy number of *alkB* gene during an oil spill simulation at mesocosm scale.

**Figure 2 microorganisms-07-00441-f002:**
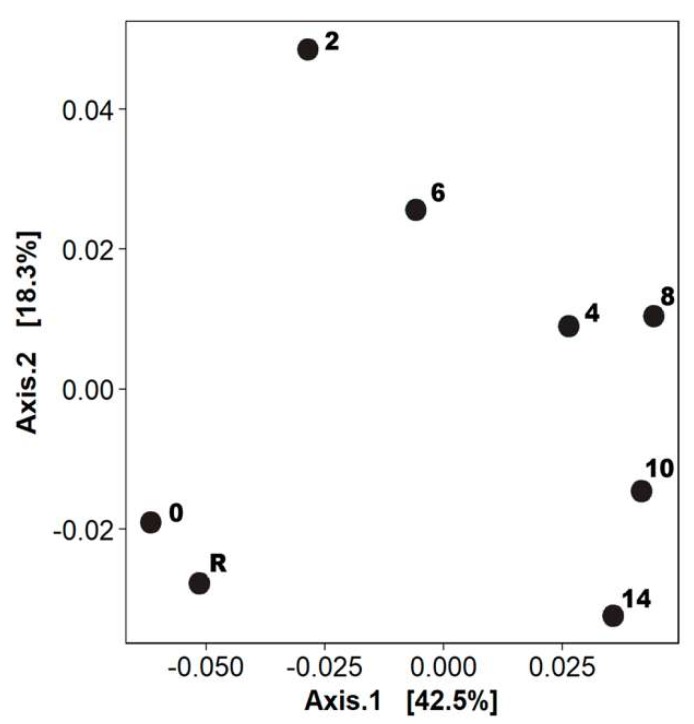
PCoA (weighted UniFrac) of bacterioplankton community during the mesocosm experiment according to amplicon sequence variants (ASVs) composition. Numbers indicate the sampling day. R = reference sample acclimatized to the mesocosm before crude oil addition.

**Figure 3 microorganisms-07-00441-f003:**
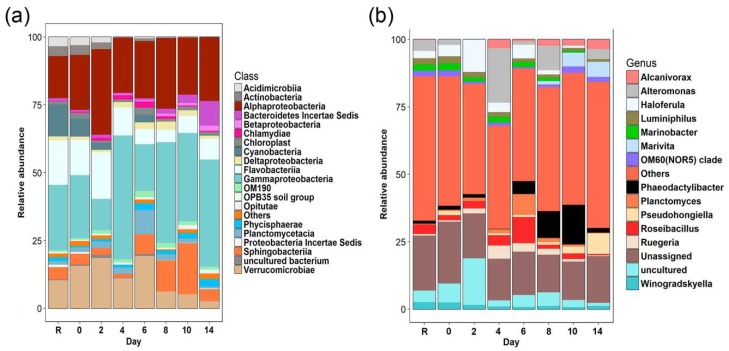
Relative abundances of the bacterioplankton community detected during the mesocosm experiment. (**a**) Classes (> 1%). (**b**) Top twenty genera.

**Figure 4 microorganisms-07-00441-f004:**
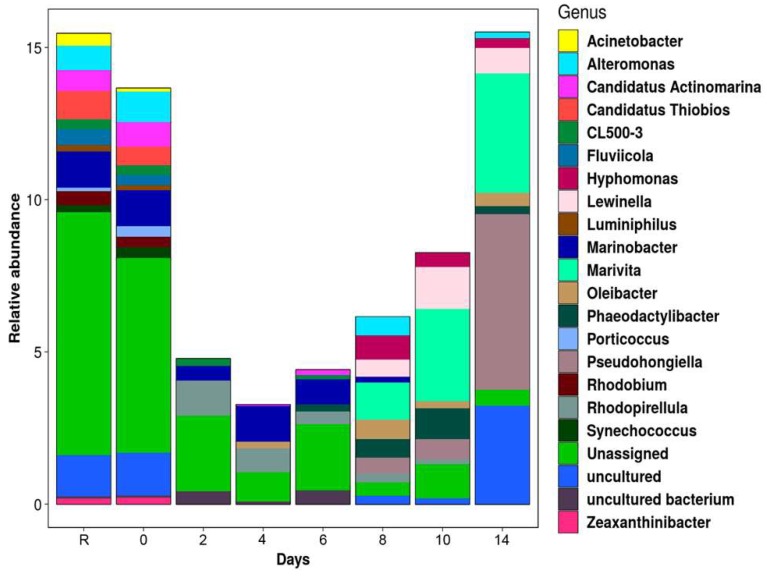
ASVs with differentiated relative abundances at genus level during the mesocosms experiment.

**Table 1 microorganisms-07-00441-t001:** Physicochemical parameters during a simulated oil spill at mesocosm scale.

Parameters	Reference	0	2	4	6	8	10	14
Total_C (%)	0.00 ±0.008	0.01 ±0.01	0.00 ±0.02	0.00 ±0.02	0.02 ±0.01	0.01 ±0.003	0.01 ±0.04	0.04 ±0.02
Total_N (%)	0.39 ±0.13	0.33 ±0.03	0.39 ±0.01	0.48 ±0.01	0.41 ±0.06	0.51 ±0.07	0.64 ±0.11	1.55 ±0.08
Temperature (°C)	28.3	27.00	26.2	27.10	26.30	26.60	27.40	27.80
O_2_ (mg L^−1^)	1.90 ±0.06	3.84 ±0.22	2.88 ±0.17	3.94 ±0.77	3.51 ±0.26	3.85 ±0.67	4.19 ±0.22	4.54 ±0.14
Salinity	36.20	36.13	35.7	35.93	35.90	35.53	36.07	36.40
PO_4_ (µmol L^−1^)	0.89 ±0.07	0.61 ±0.02	0.75 ±0.02	1.18 ±0.09	0.91 ±0.07	1.20 ±0.05	1.09 ±0.01	1.04 ±0.14
NO_2_ (µmol L^−1^)	0.08 ±0.01	0.12 ±0.002	0.10 ±0.009	0.09 ±0.09	0.10 ±0.006	0.11 ±0.01	0.07 ±0.02	0.13 ±0.001
NO_3_ (µmol L^−1^)	4.34 ±0.14	0.77 ±0.03	0.43 ±0.10	0.38 ±0.06	0.41 ±0.17	0.23 ±0.17	0.17 ±0.03	0.49 ±0.04
SiO_4_ (µmol L^−1^)	3.26 ±0.23	0.09 ±0.02	0.38 ±0.11	0.16 ±0.16	0.25 ±0.59	0.12 ±0.01	0.60 ±0.28	0.25 ±0.11
NH_4_ (µmol L^−1^)	1.04 ±0.04	1.33 ±0.01	1.42 ±0.01	0.99 ±0.01	0.83 ±0.02	0.69 ±0.005	0.76 ±0.06	1.16 ±0.51

**Table 2 microorganisms-07-00441-t002:** Alpha diversity analyses from a simulated oil spill in mesocosms experiment.

Day	Input Reads	Clean Reads	Observed ASVs	Shannon
R	150,285	45,912	263	5.18
0	133,342	39,671	236	5.02
2	115,598	32,610	178	4.66
4	113,562	34,066	196	4.74
6	126,258	37,726	231	4.94
8	135,164	39,719	234	5.08
10	145,363	42,706	255	5.06
14	293,188	64,015	317	5.08

**Table 3 microorganisms-07-00441-t003:** PERVANOVA results. The table shows the variables significatively related to changes in the bacterial community structure.

Variable	R^2^	Pr(>F)
Total Nitrogen	0.22	0.030
PAHs fraction	0.25	0.048
Naphthalene	0.32	0.003
1-metilnaphtalene	0.025	0.041
2-metilnaphtalene	0.33	0.002
Fluoranthene	0.29	0.013
Pyrene	0.28	0.026
Chrysene	0.33	0.022
Benzo(b)fluoranthene	0.29	0.013
Benzo(e)pyrene	0.30	0.025
Benzo(a)pyrene	0.35	0.005
Benzo(ghi)perylene	0.30	0.020

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
