# Peer review of "Changes in the Bacterioplankton Community Structure from Southern Gulf of Mexico During a Simulated Crude Oil Spill at Mesocosm Scale"

_microorganisms, 2019, doi:10.3390/microorganisms7100441_

Round 1

Reviewer 1 Report

General comments

In the present manuscript, the Authors investigated the possible responses of bacterioplankton to oil spills in the Southern Gulf of Mexico, using a mesocosm-based experimental approach.

This is an interesting and important topic. Please consider the following comments to improve this work. 

Abstract

In line 29 the Authors mention Alcanivorax and Alteromonas. However, from the data provided, Alcanivorax accounts for a small fraction of the relative abundance. I assume that “relative abundance” means reads amount, if so or otherwise, it should be stated in the method section and the figure captions.

In line 31 the Authors state “Nevertheless, the community structure was clearly different, with Gamma‐proteobacteria as the dominant group.” This sentence should be better rephrased to make the meaning and importance of this finding more clear. Which are the terms of this comparison? On day 2 and 6, it seems that Gamma-proteobacteria is not the dominant group. I suggest the Authors to better summarize and interpret their results in the abstract.

Introduction

The introduction provides enough information to justify the study. However, the information concerning the state of art can be improved. I would suggest to develop and provide more information concerning the in-situ consequences of oil spills. “https://doi.org/10.1146/annurev-marine-010814-015543” this may help.

Materials and methods

I don’t understand if the Reference sample belongs to the same batch used for the experiment? Could you specify?

was the thank exposed to light? Cyanobacteria completely disappear.. would light condition be responsible for this?.

Did the Authors consider/include a period of acclimatization of the mesocosms? How could this affect the experimental systems and observed results?

I could not find clear mention to the number of replicates. Based on the lack of standard deviation or standard error, I assume there is none except for RT-PCR results. Please, indicate if replicates were run and related errors.

It is not clear how the Authors used PERMANOVA to correlate (dis-)similarity matrices with environmental data (https://onlinelibrary.wiley.com/doi/full/10.1002/9781118445112.stat07841). There are several different tests to do so, possibly including BIOENV, db-RDA, and similar. Please, provide more details regarding the data matrix (replicates?) and statistical tests you used. 

Results

3.1.

Some of the environmental variables in the mesocosms are not constant between reference and treatment (for example oxygen concentrations are double or more compared with reference value). How did the Authors account for this?

3.2.

As I understand, mesocosms were supplemented with crude oil at the beginning of the experiment. Should the reader expect the highest values of TPH on the first day?

3.3.

When the Authors list the most abundant classes, they should place them in decreasing order to facilitate understanding. Moreover, Sphingobacteria (please check the spelling in figure 3a) is not mentioned although they seem to be more abundant than Cyanobacteria. There also is no mention if these estimates are means of the 8 samples.

The Authors can consider the use of SIMPER as one option to identify the role each class had on the overall (dis-)similarity (the method employed should be stated, anyway).

Authors state that according to PERMANOVA, changes in the community structure were associated with concentrations of aromatic compounds and total nitrogen. Could you please specify which test was run? Possibly, different statistical tests may be more appropriate, as commented above.

Table 2. I can see Simpson’s index. It is not mentioned anywhere. Please, consider discussing its meaning or discarding.

Discussion

Overall, I suggest the Authors provide a more balanced and fair discussion of their results and avoid over-speculation. There is a large focus on Alcanivorax, but the data provided do not support the conclusions related to this part.

4.1.

In the first paragraph, Gammaproteobacteria did not reach maximum abundance on day 2, but it did on day 4. Appropriate discussion should be provided, avoiding mere repetition of results without proper interpretation, comparison with available literature and discussion.

In particular, the changes in relative abundance of Gamma-proteobacteria should be better investigated.

The third paragraph should be improved, basing the discussion on proper statistical tests and avoiding excessive speculation.

4.2.

In the second paragraph, the Authors attribute a distinctive hydrocarbonocalastic activity to Alcanivorax on day 4. It is true that at day 4 Alcanivorax reached its highest abundance, but the data presented do not support the conclusion that it was the main responsible. Alteromonas accounted for about 20% of the relative abundance on day 4. Similarly, Ruegeria was considerably abundant. Although these two genera are discussed for their functional ability, the focus of the Authors is almost exclusively on Alcanivorax. This part of the discussion should be implemented.

Conclusions

The Authors highlight in the conclusions that several ASVs belonging to the genera Synechococcus, Acinetobacter, Candidatus Actinomarina and Candidatus Thiobios were the most affected by the presence of light crude oil throughout the experiment. However, the discussion related to this finding is not sufficient. In particular, Synechococcus is mentioned only here in the conclusion section, but never before. This should be checked and appropriate discussion and data interpretation implemented.

Minor comments

20                  SGoM, check through the text for consistency

25                  hydrocarbon degradation activity

28                  place a comma between “and” and “values.”

43                  understanding

40-41             substitute “a number of” with “several” or a synonym.

40-42             Long sentence, consider revision

49-51             please avoid the overuse of therefore, passive voice, since, in order to etc… in one single sentence.

54-58             Too long. Consider splitting into two or more sentences.

68;81              In order to -> To. Avoid the usage of in order to when possible

69-70             the aim of this study was -> this study aimed at

109                Did you use Falcon or generic brand? If the latter is correct please change to 15ml centrifuge tubes

132                 …carried out “t” in CINVESTAV-Méerida… remove the unnecessary “t”

135                paired end reads -> paired-end reads

174-176           This sentence is too long as an opening. Consider rephrasing.

184                place a period instead of a comma before “however.”

185-186            …between THE initial and final…

186                At THE genus…

188                in THE bacterial

200-201           The first sentence does not add nor introduce any subject, pros and cons as well as the application of test should be discussed in the method sections

223                “Correspondingly to this, in this work, the greatest changes” Avoid too many commas in one single sentence and rephrase. i.e. “Correspondingly to this, the greatest changes of bacterial community structure and diversity in this work were observed at day 2, in which a considerable increase in the relative abundance of Alpha‐ proteobacteria was observed (Figure 3a)”. Also, avoid the usage of “correspondingly to this” in this context. Use similarly, likewise etc…

230-234           Too long. Place a period instead of a comma here: “with Gamma‐proteobacteria as the dominant class (Figure 3a). This result agrees with…”

238-241           No statistical test is showed, I would suggest using important or considerable instead. Moreover, “it is suggested”, by who? It might be more appropriate “the result indicates that…” or similar. Also, please expand this statement providing some more information.

242-245           please rephrase this sentence which is too long and not very clear.

242-243           Please remove the comma after “compounds” if you mean the microbial species associated to the aromatic compound.

247                “The main hydrocarbon degradation activity was observed at day 4, where the highest values of culturable hydrocarbonoclastic bacteria, TPH concentration in water and AlkB number of copies were observed (Figure 1).” May be rephrased in “Hydrocarbonoclastic bacteria abundance, AlkB gene copies and TPH concentration were the highest at day 4 (Figure 1)”.

261                “…areas, however, their metabolic…”. ->”…areas, but their metabolic…”.

262                “By the end of the experiment (days 8‐14), the genus Marivita was distinctive in the ASVs differential abundance analysis (Fig 4)”. Could you explain why Marivita was “distinctive”?

263-266           The sentence could be split inasmuch as it is too long. Place a period and add a reference after aliphatic compounds.

276                …occurred after four days from disturbance

276-279           The sentence is too long and complex. Avoid using too many commas to bind long sentences. Consider rephrasing.

Author Response

General comments

Point 1: In the present manuscript, the Authors investigated the possible responses of bacterioplankton to oil spills in the Southern Gulf of Mexico, using a mesocosm-based experimental approach. This is an interesting and important topic. Please consider the following comments to improve this work. 

Response 1: We appreciate the assertive comments and constructive criticism. We have modified the manuscript according to all the comments.

Abstract

Point 2: In line 29 the Authors mention Alcanivorax and Alteromonas. However, from the data provided, Alcanivorax accounts for a small fraction of the relative abundance.

Response 2: We agree that increases in the relative abundances were best observed for Alteromonas. Thus, we have eliminated Alcanivorax from this section.

Point 3: I assume that “relative abundance” means reads amount, if so or otherwise, it should be stated in the method section and the figure captions.

Response 3: Relative abundances were presented as the number of reads converted to percentage. This information was incorporated in the methods section, lines 151 - 152. With this information incorporated in the methods we do not consider necessary to include this again in the figure captions.

“The samples were rarefied at a sample depth of 30,000 reads, then transformed to percentage to obtain relative abundances.”

Point 4: In line 31 the Authors state “Nevertheless, the community structure was clearly different, with Gamma‐proteobacteria as the dominant group.” This sentence should be better rephrased to make the meaning and importance of this finding more clear. Which are the terms of this comparison? On day 2 and 6, it seems that Gamma-proteobacteria is not the dominant group. I suggest the Authors to better summarize and interpret their results in the abstract.

Response 4: We modified this information based on the results obtained from the analysis of differential abundances for the last day of the experiment.

“At the end of the experiment, the diversity was restored to similar values as those observed in the initial time, although the community structure and composition were clearly different, where Marivita, Pseudohongiella and Oleibacter were detected to have differential abundances in days 8–14.” (Lines 29 to 32).

Introduction

Point 5: The introduction provides enough information to justify the study. However, the information concerning the state of art can be improved. I would suggest to develop and provide more information concerning the in-situ consequences of oil spills. “https://doi.org/10.1146/annurev-marine-010814-015543” this may help.

Response 5: We have provided the requested additional information in lines 48 to 58.

“In particular, for the northern GoM several studies were conducted to detect changes in the bacterioplankton community caused by the DWH disturbance. It was observed that the microbial community responded to hydrocarbons only a few weeks after the oil spill, and it was related with the long-term adaptation to natural oil released into the environment [8,9,10,11,12]. Moreover, a dominance of microorganisms related with the consumption of aliphatic hydrocarbons was found at the initial stage of the spill (e.g., Oceanospirillales and Pseudomonas), while potential degraders of aromatic compounds (e.g., Colwellia and Cycloclasticus) were observed in a second stage [8,11,13]. Towards the end of the DWH spill, another shift in the microbial community was detected, where microorganisms involved in the degradation of complex organic matter as methylotrophs, Flavobacteriia, Alteromonadaceae and Rhodobacteraceae were dominant [14].

Materials and methods

Point 6: I don’t understand if the Reference sample belongs to the same batch used for the experiment? Could you specify?

Response 6:  The reference sample belongs to the same batch used for the mesocosm. This information was included in the methods section in line 90 to 92.

“For the determination of the microbial community composition in the seawater acclimatized to the mesocosm before crude oil addition, a sample was collected and catalogued as “Reference” (R)”.

Point 7: Was the thank exposed to light? Cyanobacteria completely disappear.. would light condition be responsible for this?

Response 7: We performed an outdoor mesocosms experiment exposed to natural sunlight cycles. This information was included in the methods section (lines 88 – 89).

“The experiment was performed outdoors with natural sunlight cycles during 15 days, with constant stirring at low speed (0.125 m s-1), and no nutrient addition was performed.”

Information about the decrease in the relative abundances of Cyanobacteria was included in the discussion section (lines 240-244).

“We hypothesized that decreases in the relative abundances of Cyanobacteria could be related with the direct exposition to oil hydrocarbons, however the reduction in the amount of available light in the water column caused by the oil film on the surface, could be also associated with the affectation in the photosynthetic activity of Cyanobacteria.”

Point 8: Did the Authors consider/include a period of acclimatization of the mesocosms? How could this affect the experimental systems and observed results?

Response 8: An acclimatization period of 24 hours, prior to the oil addition was considered in our study. This information was included in the methods section (lines 86 to 88). We are aware that an acclimatization period could modify the microbial composition with respect to field site conditions. However, we decided to include this period to reduce the possibility of detecting changes in the microbial community due to the adaptation to the experimental conditions rather than those caused by the oil effect.

“An acclimatization period of 24 hours was considered in this study to allow the stabilization of the microbial communities to the experimental conditions.”

Point 9: I could not find clear mention to the number of replicates. Based on the lack of standard deviation or standard error, I assume there is none except for RT-PCR results. Please, indicate if replicates were run and related errors.

Response 9: We have included the standard deviation for measurements presented in “Table 1”. Since salinity and temperature were measured inside the mesocosm with an YSI probe, we did not perform replicates for these determinations to avoid external disturbances in the system. No standard deviations were presented for bacterial abundance because it was calculated with the most probable number method, based on the bacterial growth observed in 5 replicates, to estimate the bacterial abundance, expressed as CFU. As stated in the methods section, the hydrocarbons were quantified by GC-FID and GC-MS from 4L of water samples per day. We were not able to perform triplicates for this determination to avoid exceed the use of more than 20% of the total water in the mesocosms. It was also because other determinations, not presented in this study (e.g. phytoplankton abundance and composition), required big amounts of water samples.

Point 10: It is not clear how the Authors used PERMANOVA to correlate (dis-)similarity matrices with environmental data (https://onlinelibrary.wiley.com/doi/full/10.1002/9781118445112.stat07841). There are several different tests to do so, possibly including BIOENV, db-RDA, and similar. Please, provide more details regarding the data matrix (replicates?) and statistical tests you used.

Response 10: Pairwise dissimilarities were calculated using UniFrac (weighted) metrics [39] Each resulting dissimilarity matrix was used to visualize differences in the samples through PCoA ordination, using the “phyloseq” packages in R. The data matrix was constructed with the means of each variable per day and was used as independent variables. To test significant differences between the discriminated groups according to sampled day and physiochemical parameters, similarities were analyzed by global and pairwise PERMANOVA tests, using “adonis” function in “vegan”. P-values were obtained using 999 permutations. This information was included in the methods section lines 154 to 160.

Results

3.1.

Point 11: Some of the environmental variables in the mesocosms are not constant between reference and treatment (for example oxygen concentrations are double or more compared with reference value). How did the Authors account for this?

Response 11: We recognize that O2, NO3 and SiO4 were different in the reference respect to the rest of the days. We hypothesized that these variations could be related with the transportation conditions from field site to the facilities and their stabilization was until after 48 hours. We did not consider necessary to include a discussion of this in the manuscript because the results of the PERMANOVA indicated that these variables did not have relation with the community variation.

3.2.

Point 12: As I understand, mesocosms were supplemented with crude oil at the beginning of the experiment. Should the reader expect the highest values of TPH on the first day?

Response 12: All the results presented in this study were obtained from samples of one-meter depth in the mesocosms. Due to hydrophobicity, most of the crude oil were accumulated in the surface of the water column. Thus, the quantification of hydrocarbon corresponded exclusively to those solubilized at one-meter depth. For this reason, the highest values of TPH were observed at the fourth day, rather than the beginning of the experiment. This information was clarified in lines 97-98.

“During the experiment, one-meter depth water samples were collected from the mesocosm on days 0, 2, 4, 6, 8, 10, 14 to measure oxygen, salinity, temperature, nutrients and hydrocarbons”

3.3.

Point 13: When the Authors list the most abundant classes, they should place them in decreasing order to facilitate understanding. Moreover, Sphingobacteria (please check the spelling in figure 3a) is not mentioned although they seem to be more abundant than Cyanobacteria.

Response 13: This information was now presented in decreasing order as suggested. The Class Sphingobacteriia was included in the description of the most abundant classes, however we also specified that this information corresponded for the beginning of the experiment (day 0) where it was less abundant than Cyanobacteria (lines 190 to 192). Moreover, we verified the spelling of Sphingobacteriia and it is correct based on recently published articles (e.g., https://doi.org/10.1099/ijsem.0.002562).

“In terms of bacterial composition, the most abundant groups at the beginning of the experiment (day 0) were Gamma-proteobacteria, Alpha-proteobacteria, Verrumicrobiae, Flavobacteriia, Cyanobacteria and Sphingobacteriia (Figure 3a)”

Point 14: There also is no mention if these estimates are means of the 8 samples.

Response 14: Estimates are not the means of the 8 samples. The presented information about relative abundances corresponds to each different times of the experiments. We include more information in this paragraph for a better clarification (lines 190 to 201). 

“In terms of bacterial composition, the most abundant groups at the beginning of the experiment (day 0) were Gamma-proteobacteria, Alpha-proteobacteria, Verrumicrobiae, Flavobacteria, Cyanobacteria and Sphingobacteriia (Figure 3A). During days 2 to 4, Shannon-Weaver diversity (H’) ranged from 4.66 to 4.74 (Table 2), and Alpha- and Gamma-proteobacteria reached their maximum relative abundances, respectively (Figure 3a). At a lower taxonomical level Alteromonas increased considerably at day 4. By the last period of the experiment (days 8 to 14), the values of the diversity index were similar to those found at the initial time (Table 2). However, the final community composition was different, with notorious increases in the relative abundance of Sphingobacteriia and Bacteroidetes Incertae Sedis at the last day of the experiment. At the genus level, Marivita and Phaeodactillibacter showed an increment in their relative abundances from day 8 to 14 (Figure 3a and 3b). Genera with low relative abundance (< 3% and classified as others) represented 40-60% of the total bacterioplankton composition throughout the experiment (Figure 3b).”

Point 15: The Authors can consider the use of SIMPER as one option to identify the role each class had on the overall (dis-)similarity (the method employed should be stated, anyway).

Response 15: We used the UniFrac distance because it has some advantages over other dissimilarity measurements such as Bray-Curtis, in which SIMPER analysis is based. The UniFrac distance takes in account information on the relative relatedness of community members by incorporating phylogenetic distances between observed organisms in the computation.

We made a Kruskal Wallis test over the abundance of each ASV comparing between three groups of samples based on changes in the Shannon index. We considered it the most appropriate analysis for the detection of ASVs with differential abundances. The detailed information about this method was incorporated in lines 161 to 163.

“A Kruskal Wallis test was performed to detect ASVs with differential abundance among 3 groups of samples based on changes in the Shannon index: Group 1 (samples R and 0), Group 2 (samples 2, 4 and 6) and Group 3 (samples 8, 10 and 14).”

Point 16: Authors state that according to PERMANOVA, changes in the community structure were associated with concentrations of aromatic compounds and total nitrogen. Could you please specify which test was run? Possibly, different statistical tests may be more appropriate, as commented above.

Response 16: A permutational multivariate analysis of variance using distance Matrices was used, as above is described in a previous commentary, the metric distance was UniFrac weigthed and the test was run with “adonis” function in vegan package in R enviroment.

We consider this analysis is more robust than others since it allows the use of continue variables, besides that the analysis of variance is performed using partitional distance matrices among sources of variation and fitting linear models (e.g., factors, polynomial regression) to the distance matrices. This analysis uses a permutational test with pseudo-F ratios, therefore the determination coefficient is a pseudo R2 this R2 is based on the log likelihood for the model compared to the log likelihood for a baseline model. However, with categorical outcomes, it has a theoretical maximum value of less than 1, even for a "perfect" model. As well adjusts the scale of the statistic to cover the full range from 0 to 1.

Point 17: Table 2. I can see Simpson’s index. It is not mentioned anywhere. Please, consider discussing its meaning or discarding.

Response 17: The Simpson’s index was eliminated.

Discussion

Point 18: Overall, I suggest the Authors provide a more balanced and fair discussion of their results and avoid over-speculation.

Response 18: All the discussion was restructured according with the recommendations. Thus, a more balanced and without over-speculation discussion is now presented.

Point 20: There is a large focus on Alcanivorax, but the data provided do not support the conclusions related to this part.

Response 20: We made modifications through the document to decrease the focus on Alcanivorax. Particularly, in the discussion section it was modified the information related with this genus. Lines 269 to 274.

“During the period of highest crude oil degradation (days 2 to 4), increases in the relative abundance of Alteromonas, Haloferula, Ruegiera and Alcanivorax were distinctive (Fig 3b). These genera have been previously recognized to be involved in the degradation of aromatic and aliphatic hydrocarbons [61,62,63,64,65,66,67,68]. Therefore, we considered that these microorganisms have been related to the consumption of hydrocarbons in our experiment and may be explored for future bioremediation studies.”

4.1.

Point 21: In the first paragraph, Gammaproteobacteria did not reach maximum abundance on day 2, but it did on day 4. Appropriate discussion should be provided

Response 21: We modified this paragraph for a better understanding (234-239).

“In agreement with previous studies, significant changes in the bacterial community were observed after 48 hours of exposition to crude oil [44,45,46]. During this time, considerable increases in the relative abundances of Alpha-proteobacteria and decreases in Cyanobacteria and Gamma-proteobacteria were observed (Figure 3a). These results differ from previous works in which increases in the relative abundances of Gamma-proteobacteria and Clostridia, have been observed during the initial period of expositions to contaminants in sites with low temperature [47,48].”

Point 22: Avoiding mere repetition of results without proper interpretation, comparison with available literature and discussion.

Response 22: Several modifications were incorporated through the complete section for a better discussion.

e.g. (Lines 265-269)

“Previous mesocosms studies that have undertaken lower temperatures (≤ 18 °C) have found that the main crude oil removal occurred from 15 to 20 days [21,59]. Unlikely, in this study the highest degradation activity was observed at day 4, which highlights the importance of the seawater temperature for the hydrocarbon degradation activity as previously evidenced by Brakstad et al. [52].

Point 23: In particular, the changes in relative abundance of Gamma-proteobacteria should be better investigated.

Response 23: We modified this information to highlight that bacterial composition changed (not only Gamma-proteobacteria) at the end of the experiment despite the observed recovery in the bacterial diversity (Lines 244 - 250).

“Previous studies have reported a potential recovery of the bacterioplankton diversity in late periods of simulated hydrocarbon spills [49]. In concordance with this study, the diversity at the end of our experiment returned to similar values as those observed in the initial time. However, the bacterial composition was different from the initial community, which is also in agreement with previous studies where it has been suggested that communities do not fully recover from the disturbance and move to an alternative stable state, where they can maintain their functional capacity [7,50,51].”

Point 24: The third paragraph should be improved, basing the discussion on proper statistical tests and avoiding excessive speculation.

Response 24: We have changed the information of this paragraph to avoid over speculation in lines 256-263:

Several works have demonstrated that changes in the bacterial communities may be largely attributed to specific physicochemical parameters and hydrocarbons [53,54]. In this study, changes in the bacterioplankton structure were related to total nitrogen and the aromatic compounds concentrations, based on PERMANOVA analysis (Table 3). Accordingly, nitrogen had been related with the microbial growth and is commonly catalogued as limiting nutrient during crude oil degradation in natural environments [55]. Moreover, the persistence of aromatic hydrocarbons in the environment due to their low solubility have been well recognized to have influence on the bacterioplankton structure by previous studies [56,57,58].”

4.2.

Point 25: In the second paragraph, the Authors attribute a distinctive hydrocarbonocalastic activity to Alcanivorax on day 4. It is true that at day 4 Alcanivorax reached its highest abundance, but the data presented do not support the conclusion that it was the main responsible. Alteromonas accounted for about 20% of the relative abundance on day 4Similarly, Ruegeria was considerably abundant. Although these two genera are discussed for their functional ability, the focus of the Authors is almost exclusively on Alcanivorax. This part of the discussion should be implemented.

Response 25: Thank you so much for this important observation. We have modified this information in lines 270 to 275. Moreover, we included new information about the members putatively involved in the hydrocarbon degradation during the last days of the experiments (Lines 276 to 285).

“During the period of highest crude oil degradation (day 2 to 4), increases in the relative abundance of Alteromonas, Haloferula, Ruegiera and Alcanivorax were distinctive (Fig 3b). These genera have been previously recognized to be involved in the degradation of aromatic and aliphatic hydrocarbons [61,62,63,64,65,66,67,68]. Therefore, we consider that these microorganisms have been related to the consumption of hydrocarbons in our experiment and may be explored for future bioremediation studies.”

 “By the last stage of the experiment (days 8-14), increases in the relative abundances of Sphingobacteriia were observed (Fig 3a). Moreover, ASVs of Marivita and Oleibacter presented differential abundance in this period of the experiment (Fig 4). Previous studies have clearly showed that members of Sphingobacteriia can be n-alkane-degrading specialists [69]. Marivita belongs to the Roseobacter lineage (Alpha-proteobacteria) and has been reported to degrade aromatic and aliphatic compounds [64,70,71]. The genus Oleibacter has been reported with the capacity to degrade aliphatic hydrocarbons in tropical environments [72,73,74,75]. Based on the information presented above, we assume that Sphingobacteriia, Marivita and Oleibacter had important participation related with the hydrocarbon degradation during the last days of our mesocosm system, however further research is required for a better understanding of its contribution in the microbial community.”

Conclusions

Point 26: The Authors highlight in the conclusions that several ASVs belonging to the genera SynechococcusAcinetobacter, Candidatus Actinomarina and Candidatus Thiobios were the most affected by the presence of light crude oil throughout the experiment. However, the discussion related to this finding is not sufficient. In particular, Synechococcus is mentioned only here in the conclusion section, but never before. This should be checked and appropriate discussion and data interpretation implemented.

Response 26: We have incorporated information about this in the discussion section (Lines 251-255) and we modified the conclusions (289 to 302)

“In our study, some of the main changes in the bacterial composition after the simulated oil spill were related with decreases in the relative abundances of Candidatus Thiobio, Candidatus Actinomarina, Acinetobacter and Synechococcus. Previous studies have reported to Synechococcus and Actinomarina as sensitive taxa for contaminated sites with heavy metals [52]. However, futures studies are needed in order to understand the oil effect over these genera.”

“The mesocosm experiment of this study evidenced a decrease in bacterioplankton diversity and changes in the community composition from sGoM after two days from the simulated oil spill. Increases in the relative abundances of Alpha-proteobacteria and decreases in Cyanobacteria and Gamma-proteobacteria were some of the main changes in the bacterial composition observed in this period of the experiment. According to the TPH concentration, counts of culturable hydrocarbonoclastic bacteria and alkB gene copy number, the greatest hydrocarbon degradation activity performed by the bacterioplankton of sGoM occurred after four days from the disturbance. Alteromonas, Haloferula, Ruegiera and Alcanivorax were found as the main groups putatively involved in the hydrocarbon degradation, based on increases in their relative abundances at this point of the experiment. Despite of the bacterial diversity being recovered at the end of the experiment (14 days), the community composition was different from the initial time. To our knowledge, this is the first mesocosm experiment to evaluate the effect of oil spills over the bacterioplankton community from sGoM. All the results presented here can be employed as a guideline for future studies aiming to evaluate bioremediation strategies for marine environments.”

Point 27: Minor comments 

Response 27: We made all the suggested modifications. All these changes were highlighted in the Word document in track mode.

Point 28:20 SGoM, check through the text for consistency  

Response 28: We have changed SGoM to sGoM through the complete document. (lines 20, 35, 45, 72, 76, 77, 228, 234, 290 and 300)

Point 29: 25 hydrocarbon degradation activity

Response 29: We have made this modification. (line 25)

Point 30: 28 place a comma between “and” and “values.”

Response 30: We have made this modification. (line 28)

Point 31: 34 understanding

Response 31: We modified this statement in line 35:

“…results of this study contribute to the understanding of the…”

Point 32: 40-41 substitute “a number of” with “several” or a synonym.

Response 32: This change was included (line 40)

Point 33: 40-42 Long sentence, consider revision

Response 33: We have modified the sentence in lines 40-41:

“The Gulf of Mexico (GoM) is a hotspot of biological diversity in which several oil-related industries are hosted, representing a high risk for oil spill disasters.”

Point 34: 49-51 please avoid the overuse of therefore, passive voice, since, in order to etc… in one single sentence.

Response 34: We have modified the sentence in lines 59-60:

“For over a decade, several simulations at mesocosm scale have been performed for the evaluation of the effect of hydrocarbons on planktonic microorganisms.

Point 35: 54-58 Too long. Consider splitting into two or more sentences.

Response 35: We have modified this sentence in lines 64-66:

“Moreover, previous investigations at mesocosms scale have reported Alcanivorax as relevant microorganisms for cleaning oil-polluted marine systems, due to their high capability to use aliphatic compounds as carbon source [19,20,21].

Point 36: 68;81 In order to -> To. Avoid the usage of in order to when possible

Response 36: We have made the proper modifications in lines 71-77 and 89-91.

Thus, contributing with the understanding of the potential response of the bacterioplankton from sGoM to crude oil spills, this study aimed to assess the effect of light crude oil over the native microbial community from sGoM, through a simulated oil spill under a mesocosm experiment.

For the determination of the microbial community composition in the seawater acclimatized to the mesocosm before crude oil addition, a sample was collected and catalogued as “Reference” (R)

Point 37: 69-70 the aim of this study was -> this study aimed at

Response 37: We have made this modification in line 76.

Point 38: 109 Did you use Falcon or generic brand? If the latter is correct please change to 15ml centrifuge tubes

Response 38: Since a generic brand was used, we have changed the description of “15 mL Falcon tubes” for “15 mL centrifugate tubes) in line 119.

Point 39: 132 …carried out “t” in CINVESTAV-Mérida… remove the unnecessary “t”

Response 39: This correction has been made in line 142.

Point 40: 135 paired end reads -> paired-end reads

Response 40: This correction has been made in line 145.

Point 41: 174-176 This sentence is too long as an opening. Consider rephrasing.

Response 41: We made this modification on lines 188 to 190:

“Initial bacterioplankton community displayed the highest values of Shannon-Weaver index (Table 2). PCoA indicated that community structure after the oil addition was highly distant from the initial time (Figure 2).

Point 42:184 place a period instead of a comma before “however.”

Response 42: This modification was made in line 196.

Point 43:185-186 …between THE initial and final…

Response 43: We have made this modification in line 197.

Point 44: 186 At THE genus…

Response 44: We have made this modification in line 199.

Point 45:188 in THE bacterial

Response 45: The paragraph change to “However, the final community composition was different, with notorious increases in the relative abundance of Sphingobacteriia and Bacteroidetes Incertae Sedis at the last day of the experiment. At the genus level, Marivita and Phaeodactillibacter showed an increment in their relative abundances from day 8 to 14 (Figure 3a and 3b).” (lines 195 to 199).

Point 46: 200-201 The first sentence does not add nor introduce any subject, pros and cons as well as the application of test should be discussed in the method sections

Response 46: We have eliminated this sentence and the information was presented in the methods section in lines 215-216.

Point 47: 223 “Correspondingly to this, in this work, the greatest changes” Avoid too many commas in one single sentence and rephrase. i.e. “Correspondingly to this, the greatest changes of bacterial community structure and diversity in this work were observed at day 2, in which a considerable increase in the relative abundance of Alpha‐proteobacteria was observed (Figure 3a)”. Also, avoid the usage of “correspondingly to this” in this context. Use similarly, likewise etc…

Response 47: we modified this information on lines 236 to 245.

Point 48: 230-234 Too long. Place a period instead of a comma here: “with Gamma‐proteobacteria as the dominant class (Figure 3a). This result agrees with…”

Response 48: we modified this information on lines 246-251

Point 49: 238-241 No statistical test is showed, I would suggest using important or considerable instead. Moreover, “it is suggested”, by who? It might be more appropriate “the result indicates that…” or similar. Also, please expand this statement providing some more information.

Response 49: we modified this information on lines 257 to 264

Point 50: 242-245 please rephrase this sentence which is too long and not very clear.

Response 50 : we modified this information on lines 257 to 264

Point 51: 242-243 Please remove the comma after “compounds” if you mean the microbial species associated to the aromatic compound.

Response 51: we modified this information on lines 257 to 264

Point 52: 247 “The main hydrocarbon degradation activity was observed at day 4, where the highest values of culturable hydrocarbonoclastic bacteria, TPH concentration in water and AlkB number of copies were observed (Figure 1).” May be rephrased in “Hydrocarbonoclastic bacteria abundance, AlkB gene copies and TPH concentration were the highest at day 4 (Figure 1)”.

Response 52: we modified this information on lines 266 to 270

Point 53: 261 “…areas, however, their metabolic…”. ->”…areas, but their metabolic…”.

Response 53: we modified this information on lines 271 to 276.

Point 54: 262 “By the end of the experiment (days 8‐14), the genus Marivita was distinctive in the ASVs differential abundance analysis (Fig 4)”. Could you explain why Marivita was “distinctive”?

Response 54: we modified this information on lines 278 to 287

Point 55: 263-266 The sentence could be split inasmuch as it is too long. Place a period and add a reference after aliphatic compounds.

Response 55: we modified this information on 281 to 284

“Marivita belongs to the Roseobacter lineage (Alpha-proteobacteria) and has been reported to degrade aromatic and aliphatic compounds [64,70,71]. The genus Oleibacter has been reported with the capacity to degrade aliphatic hydrocarbons in tropical environments [72,73,74,75]. “

Point 56: 276 …occurred after four days from disturbance

Response 56: This modification was made in line 352.

Point 57: 276-279 The sentence is too long and complex. Avoid using too many commas to bind long sentences. Consider rephrasing.

Response 57: We have modified this sentence according to the recommendations in lines 352-359.

Reviewer 2 Report

In the present work, the authors investigate the response of natural microbial population trying in superficial water in the Southern Gulf of Mexico (sGoM) to a simulated oil spill using a mesocosm approach.
The general experimental procedure was well arranged and explained in the materials and methods,
although the authors limited their attention to 16s-base DNA microbial biodiversity analysis.
The presence and abundance of the alkB gene was the only investigation of the “functional potential” of the microbial community.
The only use of DNA instead of RNA (that could be more useful to distinguish between active and not-active microbial communities) contribute to decreasing the "novelty" of the results obtained.
The only interesting results were discussed in the discussion: (line 221-227) where the temperature was found to have an important role to discriminate the succession of oil-degrading bacteria and the line (262-268) were the genus Marivita was found important as key player in the ASVs differential abundance analysis and it's potential as degrader of aromatic and aliphatic compounds.
Given the layout of the work and the presentation of the results, the work can be Accept in present form.

Author Response

Point 1: In the present work, the authors investigate the response of natural microbial population trying in superficial water in the Southern Gulf of Mexico (sGoM) to a simulated oil spill using a mesocosm approach. The general experimental procedure was well arranged and explained in the materials and methods, although the authors limited their attention to 16s-base DNA microbial biodiversity analysis. The presence and abundance of the alkB gene was the only investigation of the “functional potential” of the microbial community. The only use of DNA instead of RNA (that could be more useful to distinguish between active and not-active microbial communities) contribute to decreasing the "novelty" of the results obtained. The only interesting results were discussed in the discussion: (line 221-227) where the temperature was found to have an important role to discriminate the succession of oil-degrading bacteria and the line (262-268) were the genus Marivita was found important as key player in the ASVs differential abundance analysis and it's potential as degrader of aromatic and aliphatic compounds. Given the layout of the work and the presentation of the results, the work can be Accept in present form.

Response 1: We thank for the positive comments and remarks upon of this work. Unfortunately, for this study we did not consider RNA analyses, however we agree that the use of RNA for community analyses is most suitable for the detection of active bacteria. This will be considered for further studies.

Reviewer 3 Report

This is a highly topical and interesting study, especially in light of the ongoing industrialization of the GoM. I have not many negative comments, besides that the grammar needs a slight touch-up. A good study, which could be used as fundamental research on the impact of oil spills on planktonic communities in the Caribbean.

Author Response

Point : This is a highly topical and interesting study, especially in light of the ongoing industrialization of the GoM. I have not many negative comments, besides that the grammar needs a slight touch-up. A good study, which could be used as fundamental research on the impact of oil spills on planktonic communities in the Caribbean.

Response : We appreciate the interest in our study and then positive comments. According to the recommendation, the manuscript has now been revised thoroughly about language editing by an expert company.

Round 2

Reviewer 1 Report

The Authors have accomplished the requests made. I endorse publication of the revised version of this manuscript.